# Friction and Wear Behavior between Crane Wire Rope and Pulley under Different Contact Loads

Xiangdong Chang [1,2], Xiao Chen [3,4,*], Yaoyuan Dong [5], Hao Lu [1,2], Wei Tang [1,2], Qing Zhang [1,2] and Kun Huang [1,2]

[1] Jiangsu Key Laboratory of Mine Mechanical and Electrical Equipment, School of Mechanical and Electrical Engineering, China University of Mining and Technology, Xuzhou 221116, China
[2] Jiangsu Collaborative Innovation Center of Intelligent Mining Equipment, Xuzhou 221116, China
[3] State Key Laboratory of Mining Response and Disaster Prevention and Control in Deep Coal Mines, Anhui University of Science and Technology, Huainan 232001, China
[4] Coal Mine Safety Mining Equipment Innovation Center of Anhui Province, Anhui University of Science and Technology, Huainan 232001, China
[5] Angang Steel Company Limited, Anshan 114000, China
[*] Correspondence: 2020130@aust.edu.cn; Tel.: +86-150-6219-5713

**Abstract:** Surface wear caused by contact between crane wire rope and a pulley seriously affects the mechanical properties of the wire rope. In this study, the tribological behavior of wire rope was investigated using a homemade rope–pulley sliding friction test rig. Then, the influence of different surface wear on the bending fatigue life of the rope samples was analyzed. The results show that the friction coefficient (COF) decreases with the increasing sliding distance. It reaches a minimum of approximately 0.52 when the contact load is 700 N. The surface temperature of the wire rope rises rapidly and then gradually stabilizes. The maximum temperature rise fluctuates in the range of 50 °C to 60 °C with increasing contact load. The wear scar of the wire rope is irregular, and the maximum wear width increases from approximately 1.94 mm to 2.45 mm with the contact load. Additionally, increased contact load leads to smoother wear surface of wire rope, and the wear mechanisms are mainly abrasive wear and adhesive wear. Additionally, surface wear leads to a decrease in the bending fatigue life of wire ropes, and degradation of anti-bending fatigue is more serious under a larger sliding contact load.

**Keywords:** crane wire rope; pulley groove; sliding friction; surface wear; bending fatigue

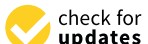



## 1. Introduction

Wire rope is an important bearing and transmission component of crane machinery. The mechanical properties of the wire rope in service are directly related to the safe and reliable operation of the equipment system and the life safety of operators [1]. However, during operation of the crane, the wire rope is affected by the system structure and working conditions, resulting in different wear of the wire rope [2,3]. This in turn leads to its performance degradation and service life reduction. In order to achieve a wide range of heavy lifting, wire rope is used with pulleys, as shown in Figure 1. When the crane completes hoisting, rotating, amplitude variation and driving functions, the wire rope is subjected to irregular vibration and impact loads [4,5]. This leads to relative sliding and frictional wear between the wire rope and the pulley groove [6,7]. Additionally, the wire rope–pulley system is in a suspended state during service, and it will shake under loading conditions. Thus, the wire rope will twist and swing on the pulley groove, resulting in complex sliding contacts and severe surface wear. In this engineering application, the tribological behavior between the wire rope and the pulley not only harms the service performance of wire ropes but also affects the operation stability of the hoisting system. Therefore, exploring the sliding friction characteristics and wear mechanism between

the wire rope and the pulley is of great significance to its structural design, safe use and maintenance.

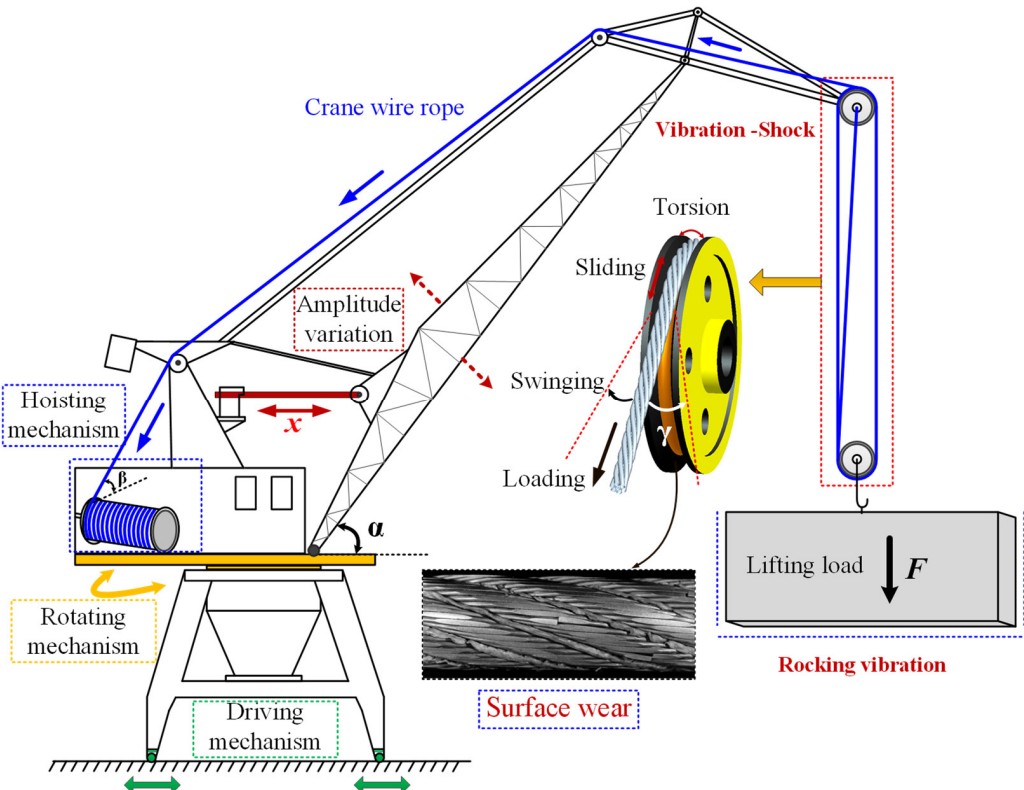

**Figure 1.** Schematic diagram of operation system of crane wire rope.

In recent years, many scholars have carried out research on damage and safe service of wire rope under actual working conditions. Wear, corrosion and fatigue are the main damage forms of wire ropes [8]. To explore the fretting friction and wear behavior between internal steel wires of wire rope caused by complex stress state, Wang et al. [9,10] investigated the dynamic wear evolution and fretting fatigue characteristics between steel wires under different sliding amplitudes. Larger relative displacement reduces the surface wear of steel wires. Cruzado et al. [11–13] deeply analyzed the influence of contact load and crossing angle on fretting friction and wear between steel wires by simulation and experiment. The results show that the wear volume caused by fretting between steel wires depends mainly on variation in normal contact load. Additionally, considering the influence of the structural characteristics of wire rope, different spiral contact states between the steel wires will also affect their fretting fatigue behavior [14]. Furthermore, because the service environment of wire rope is very harsh, corrosive environmental media have a greater impact on the surface lubrication and friction state of the wire rope [15]. Wang et al. [16] analyzed the tribo-fatigue behaviors of rope wires under different environmental media (air, acid, neutral and alkaline electrolyte solutions, and deionized water). They found that the wear mechanisms include abrasive wear, adhesive wear, corrosion wear and fatigue wear. Sun et al. [17] carried out an experimental study on modification of wire rope grease and found that the anti-wear ability of the base grease can be enhanced by addition of multilayer graphene or micron graphite. In addition to fretting wear between internal wires, surface wear of wire ropes due to external contact is also an important cause of damage and performance degradation of wire ropes. Considering the influence of vibration shock and ambient temperature, Peng et al. [18,19] explored the sliding friction and wear characteristics of wire ropes under different impact loads and low-temperature environments. They found that effective lubrication reduces impact vibration and low temperatures increase the COF between wire ropes. Oksanen et al. [20,21] analyzed the

contact and wear characteristics between wire rope and a pulley, focusing on the effect of relative sliding on the surface damage of the groove. The results show that material removal proceeds through spalling of the deformation tongues by crack growth between the graphite nodules and the surface. Moreover, the main harm of wear to wire rope is degradation of its mechanical properties, which in turn reduces the service life of wire ropes and threatens the safety and reliability of the equipment system. Chang et al. [22] explored the influence of different wear scars on the breaking strength of wire rope. They found that surface wear causes breaking elongation of wire rope to decrease and changes the fracture morphology. Jikal et al. [23] studied the influence of different corrosions on the fatigue limit of wire ropes through comparative tests and proposed a method for predicting the life of wire ropes in corrosive environments. They found that corrosion could lead to a significant decrease in fatigue strength and accelerate fatigue damage of corroded steel strands. Fatigue life decreases linearly with corrosion level. Chen et al. [24] established a corrosion damage model of multi-layer strand wire rope and analyzed the effect of pitting corrosion on its mechanical properties. Wahid et al. [25] studied the effect of number of broken wires on fracture failure of wire ropes and proposed a damage prediction method. By analyzing the mechanical and chemical properties of wire rope, they found that the behavior of wire rope is semi-brittle. Based on a bending fatigue test of wire ropes, Battini et al. [26] proposed a thermal method for estimating fatigue life of metallic ropes. The proposed model showed a very good correlation between early-stage data and first wire failure conditions. Then, the research results realized evaluation of damage and life of wire ropes by monitoring heat changes.

To improve the tribological behavior of wire rope under different service conditions, Zhang et al. [27,28] studied the tribological behavior of wire rope under the condition of modified lubricating oil and analyzed the effect of different additives on reducing the surface wear. They found that lanthanum-stearate-modified lubricating oil can better reduce the wear of wire rope under different sliding speeds and contact loads. Chang et al. [29] studied the tribological properties of lubricated wire ropes in different corrosive environments. The results show that the corrosion solutions degenerate the anti-friction and anti-wear properties of the lubricating grease and oil. McColl et al. [30] studied the friction and wear properties of steel wire under different lubrication conditions and found that grease lubrication can form a better protective layer on the contact surface and reduce the COF. Périer et al. [31] analyzed the fretting friction and wear behaviors of wires in NaCl solution and aqueous solution. The results show that the influence of NaCl solution on fretting fatigue life is not obvious. Molnár et al. [32] studied the performance degradation of wire rope and internal wire in salt solution. It is helpful to predict the service life of wire ropes. Wu et al. [33] studied the influence of sulfide concentration, stress level and pH value on stress corrosion cracking of steel wire. They found that galvanized coating was effective in reducing the corrosion.

Through the above analysis, it can be found that the existing research mainly focuses on fretting fatigue of steel wires and surface wear between wire ropes. However, the variation characteristics of friction behavior and wear mechanisms mainly depend on tribological (friction) pairs [34,35]. There are few studies on the rope–pulley friction and wear system. This surface wear is also common under service conditions of crane wire ropes. Additionally, it is difficult to accurately judge the service state and safety reliability of wire rope without fully mastering the damage forms and characteristic mechanisms. Therefore, it is necessary to study the tribological behavior of crane wire rope in the process of bending over the pulley and reveal the damage to its service performance.

In this paper, sliding friction and wear tests between wire rope and a pulley under different contact loads were carried out. Variations in the COF, friction temperature rise and characteristic parameters (wear width and wear loss) of surface wear under different test conditions were measured and analyzed. Additionally, the characteristics and wear mechanisms of the wear surface were revealed. Finally, the fracture failure behaviors of different worn wire ropes under bending fatigue load were analyzed. The degradation

characteristics of the bending fatigue life of the rope samples were studied. Related research can reveal the rope–pulley friction characteristics and establish the relationship between wear and bending fatigue life degradation. The research results can provide an important basis for prolonging the service life of wire rope and guiding its safe use.

## 2. Materials and Methods

### 2.1. Test Sample

In this paper, the friction pairs include wire rope and pulley, as shown in Figure 2. The pulley is made of Q235 steel plate (see Figure 2b,c). Its chemical composition (in wt%) is 0.18 C, 0.46 Mn, 0.275 Si, 0.045 S and 0.04 P. Additionally, the diameter of the pulley is 400 mm, and the diameter of the rope groove is 14 mm. Furthermore, the structure of the wire rope is presented in Figure 2d,e. It is 6 × 19 galvanized wire rope with fiber core (SZ-B-0.60-YB/T5294-2009), which contains 6 strands, and each strand consists of 19 steel wires. The internal steel wires are manufactured by the cold drawing process from high-quality carbon structural steel. Additionally, the thickness of galvanized layer of the steel wire is approximately 7 μm to 20 μm. The density of galvanized low carbon steel wire is approximately $7.81 \times 10^3$ kg/m$^3$. The surface morphology of steel wires is presented in Figure 2f. The chemical composition of the wire material (in wt%) is 98.69 Fe, 0.87 C, 0.39 Si, 0.03 Mn and 0.01 Ni, and the other elements include S and P. The detailed structural parameters of the wire rope are shown in Table 1.

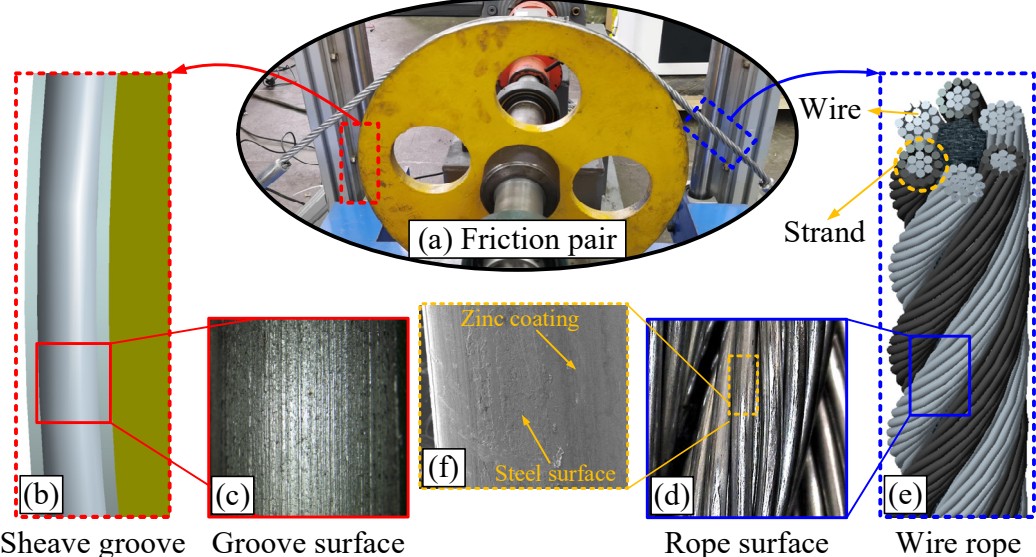

**Figure 2.** Surface structure of friction pairs: (**a**) contact form, (**b**) and (**c**) pulley, (**d**) and (**e**) wire rope, (**f**) surface morphology of steel wires.

**Table 1.** Structure parameters of the wire rope.

| Parameter | Value |
|---|---|
| Length of the rope sample (mm) | 600 |
| Diameter of the rope (mm) | 9.3 |
| Radius of the steel wires (mm) | 0.3 |
| Strand lay length (mm) | 70 |
| Strand lay angle (°) | 15.5 |
| Strand lay direction | Right |
| Nominal tensile strength (Mpa) | 1570 |
| Breaking force (N) | 52,500 |

### 2.2. Test Procedure

To investigate the tribological behavior between wire rope and pulley and the effect of surface wear on its bending fatigue life, the sliding friction and wear tests were conducted using a homemade test rig, as shown in Figure 3a. The contact angle of the wire rope is 45°. Then, the bending fatigue tests of the damaged rope samples were conducted using a customized testing machine, as shown in Figure 3d. The friction test rig mainly includes drive system, rotation system, loading system and data acquisition system. The pulley is connected with variable frequency motor through a drive shaft. It can realize continuous rotation under the control of frequency converter. Both ends of the wire rope are connected with the floating loading device, and the wire rope is in contact with the pulley rope groove (see Figure 3b,c). During the wear test, the wire rope is clamped firmly at both ends. Additionally, the floating loading device is connected to the linear guide rails by four sliders, allowing it to move freely in the vertical direction. Therefore, the contact load between the wire rope and the pulley is the weight of the floating loading device and the wire rope. Furthermore, the contact load of the friction pair can be adjusted by placing different heavy blocks on the floating loading device. Considering that the crane wire rope is mostly in service under variable loading conditions, sliding friction and wear tests of the wire rope under different contact loads ($F$) were conducted. The sliding friction torque of the wire rope can be monitored in real time by a dynamic torque sensor (between pulley shaft and the motor) and data acquisition system. The bending fatigue testing machine can realize continuous left and right bending of the damaged rope samples, as shown in Figure 3d. One end of the wire rope is connected to the rocker and the other end is fixed by the arc fixture (see Figure 3e). During the bending fatigue test, the contact state between the wire rope and the arc fixture is shown in Figure 3f. Additionally, the controller of the testing machine can adjust the bending frequency of the wire rope and display the bending times in real time. Therefore, the influence of surface wear on the bending fatigue life of wire ropes can be revealed by analyzing the relationship between the number of broken wires and the bending fatigue times of different damaged ropes. In most engineering applications, wire ropes are judged to be scrapped when the wire breakage rate reaches 10%. The bending fatigue times of the first 12 broken wires were studied in this paper. Moreover, the detailed parameters of the tribological test and the bending fatigue test are shown in Table 2.

### 2.3. Test Parameters and Methods

The sliding friction characteristics between the wire rope and the pulley were analyzed by the COF and the temperature rise (see Figure 4a). The friction force of the wire rope can be obtained by the friction torque measured by the dynamic torque sensor and the radius of the pulley. Then, combined with the contact load between the wire rope and the pulley, the COF of the wire rope can be calculated. Therefore, this paper analyzed the variation in the COF with sliding distance and contact load. During the sliding friction test, the temperature distribution characteristics of the wire rope were monitored and recorded by an infrared thermal imager. It enables real-time detection and data recording of the surface temperature of objects in the observation area. Then, the variation temperature rise in the contact region was analyzed through the maximum temperature and the room temperature (see Figure 4a). Additionally, the distribution of the surface wear scar was observed using an optical microscope. The maximum wear width of the wear scar was measured, as shown in Figure 4b,c. The structural contour of the wire rope is distinguished by the color change in the 3D diagram, and the same color corresponds to the same height. The redder the color, the greater the relative height. Furthermore, the weight of wear debris produced by the friction pair during the wear test was measured by an electronic balance. The measurement accuracy is 0.1 mg. The wear characteristics and wear mechanisms of the crane wire rope were analyzed by the optical microscope and scanning electron microscope (SEM), as shown in Figure 4d. Finally, the variation law of broken wires and bending fatigue times of the rope sample were analyzed (see Figure 4e). The distribution

characteristics and fracture morphology of broken wires in the wear region were observed and discussed.

**Table 2.** Experimental parameters.

| Sliding Friction and Wear Test | | Bending Fatigue Test | |
|---|---|---|---|
| Parameters | Values | Parameters | Values |
| Contact load (N) | 600, 650, 700, 750, 800 | Sample length (mm) | 350 |
| Sliding distance (m) | 240 | Frequency (Hz) | 1 |
| Sliding velocity (m/s) | 0.8 | Number of broken wires | 12 |
| Contact angle (°) | 45 | Bending radius (mm) | 100 |
| Contact arc length (mm) | 157 | | |
| Ambient temperature (°C) | 25 ± 5 | | |
| Relative humidity (%) | 50 ± 10 | | |
| Atmosphere | Laboratory air | | |

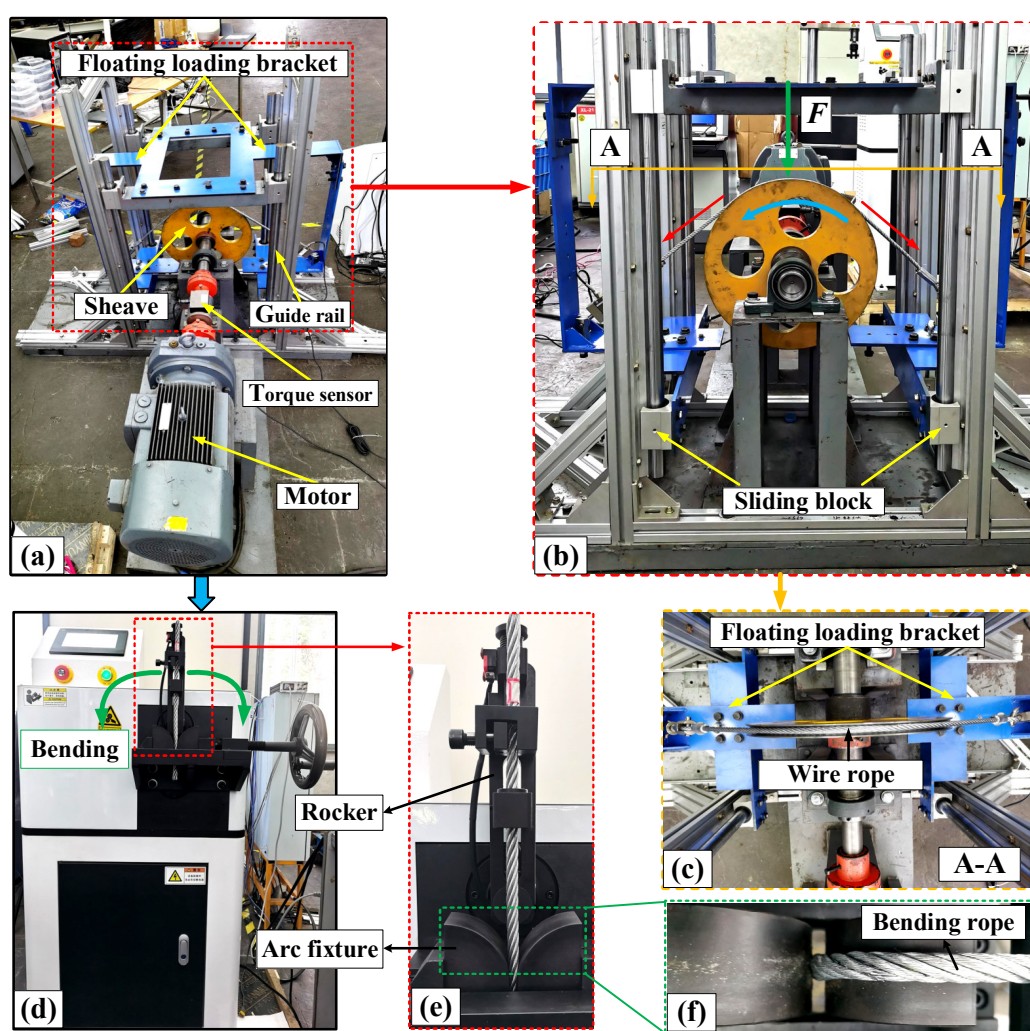

**Figure 3.** Structure of test rigs: (**a**) sliding friction and wear test rig, (**b**) front view of friction tester, (**c**) top view of the friction tester, (**d**) breaking tensile testing machine, (**e**) tensioning and fixing of wire rope, (**f**) bending contact state of wire rope.

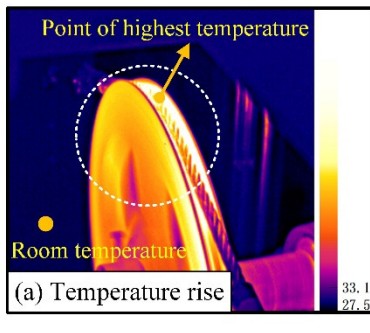
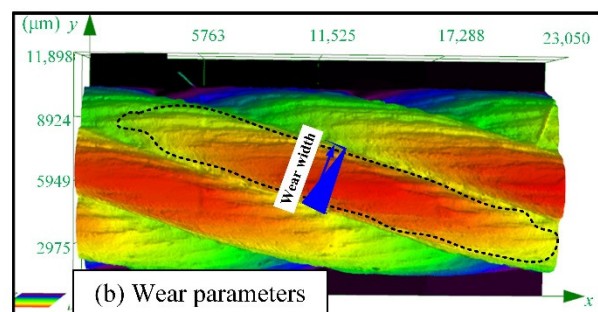

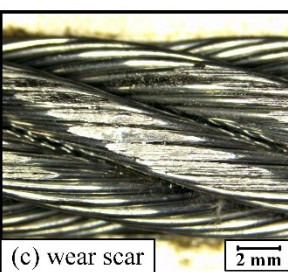
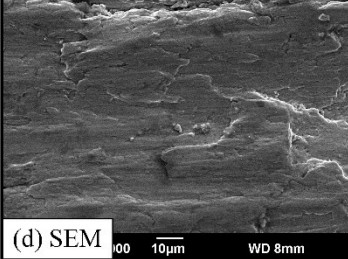
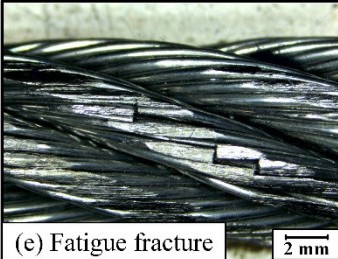

**Figure 4.** Main research parameters of tribological properties and bending fatigue behavior: (**a**) infrared thermal image, (**b**) 3D topography of wear scar, (**c**) optical micrograph of wear scar, (**d**) SEM of wear surface, (**e**) fatigue broken wires.

## 3. Results and Discussion

### 3.1. Sliding Friction Parameters

Figure 5 shows the variation in the COF between wire rope and a pulley under different contact loads. Because the surface structure of the wire rope is complex and is constantly changing during the sliding wear, the variation trend of the COF with increasing sliding distance is obvious (see Figure 5a). In the initial stage of the sliding friction test, the spiral wires of the rope surface contact with the pulley groove directly. The surface of the friction pair is mainly line contact, and the contact area is small. Thus, the COF decreases rapidly from the larger position when the sliding distance is less than approximately 30 m. Additionally, as the structures of the wire rope and steel wires are both cylindrical, the contact area of the friction pair will continue to increase during the sliding process, and the increase rate is fast and then slow. The contact form changes from line contact to surface contact and the contact stress decreases. The COF shows a decreasing trend and its decreasing speed becomes significantly smaller as the sliding distance increases from approximately 50 m to 180 m. Finally, when the sliding distance exceeds 180 m, the COF of the wire rope enters a relatively stable stage. This indicates that the contact state between the wire rope and the pulley is relatively stable, and the increase in the sliding distance does not have a great influence on the surface characteristics. Therefore, the COF between the crane wire rope and the pulley is constantly changing in the process of service. The wear state of the wire rope can be judged by its change rule. Moreover, under different contact loads, the variation trend of the COF with sliding distance is basically the same, but there are some differences in the stable stage. Figure 5b shows the average COF in the relative stable stage under different contact loads. It is clear that the effect of the contact load on the COF is not obvious. The COF fluctuates from approximately 0.5 to 0.6 as the contact load increases from 600 N to 800 N. Additionally, with the contact load increasing from 600 N to 700 N, the COF decreases from 0.58 to 0.52. Then, with the contact load continuing to increase to 800 N, the COF increases linearly to 0.56. This means that the COF of the wire rope is the minimum at the contact load of 700 N. The variation in the COF is due to the change in the surface characteristics of the wire rope under different contact loads. Additionally, the structure of wire rope is complex, and the change in its contact state during the experiment will also have a great influence on the COF [36]. The increase

in contact load will increase the effective contact area of the metal friction pair, resulting in an increase in the COF. However, during the sliding process, the larger load is more likely to cause plastic deformation of the micro-protrusions on the contact surface, which in turn causes the surface pits and protrusions to complement each other. This ultimately results in a smoother sliding contact surface of the wire rope. The COF under this sliding condition is relatively smaller. Therefore, in the process of increasing the contact load, the two COF variation characteristics are balanced with each other. The COF will have a minimum value. This shows that the smooth surface of the wire rope plays a leading role in the COF when the contact load is 700 N. Furthermore, as the contact load continues to increase, the adhesion force increases, and the COF increases during the sliding process. Moreover, the contact load mainly affects the surface wear degree of the wire rope, and the influence on the COF in the stable stage is relatively small. The fluctuation of the COF with contact load is mainly determined by an increase in the temperature in the friction region and the surface wear characteristics of wear scars.

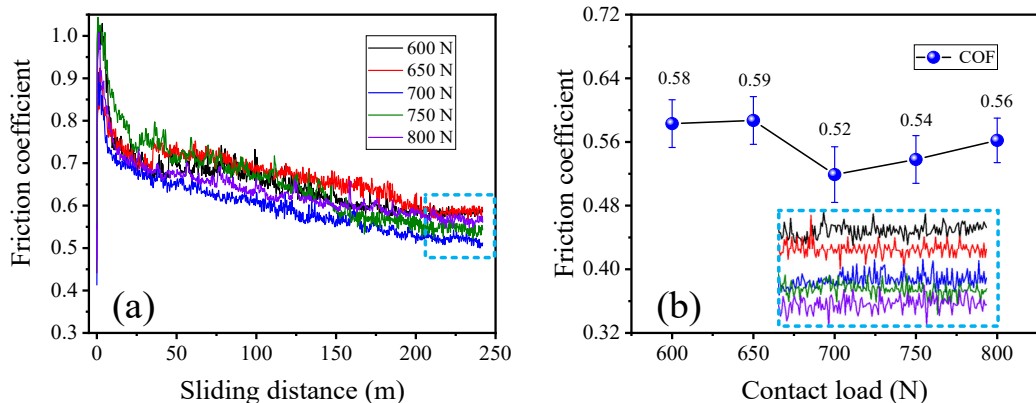

**Figure 5.** COF curves between wire rope and pulley under different sliding conditions: (**a**) process variation curves with sliding distance, (**b**) average COF in the relatively stable stage.

The infrared thermograms of a friction pair under different contact loads are shown in Figure 6. The surface color of the contact area between the wire rope and the pulley is significantly brighter. This means the temperature in the region is higher and a great deal of frictional heat is generated during the friction test. Additionally, the temperature distribution characteristics caused by the continuous sliding between the wire rope and the pulley can be visualized through the infrared thermal image. The bright area spreads from the top of the contact arc between the rope and the groove to the surrounding area. Since the pulley rotates continuously during the test, the temperature of its circumference is the highest, and, the closer to the center of the circle, the lower the temperature. Furthermore, because the wire rope is a complex structure made of many twisted steel wires, its thermal conductivity is relatively poor. The color difference between the contact area and the non-contact area of the wire rope is obvious. This indicates that sliding friction wear only has a significant effect on the temperature of the contact area of the wire rope. Therefore, in the service process of crane wire rope, the temperature rise of the wire rope is a transient process and its impact is not sustainable. However, sliding friction can cause the temperature of the pulley to remain at a high level. Moreover, an increase in contact load will lead to a change in the surface temperature of the wire rope, but it is difficult to distinguish through the color of the infrared thermal images.

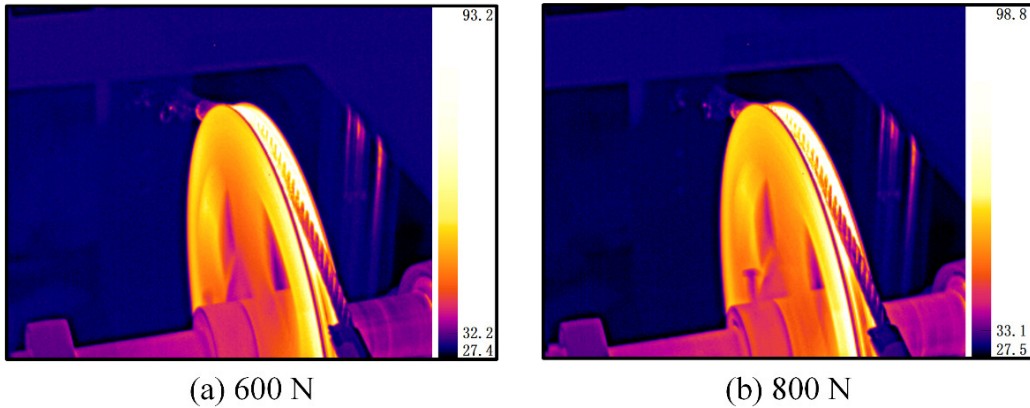

(a) 600 N
(b) 800 N

**Figure 6.** Infrared thermal images of the friction pairs: (**a**) contact load of 600 N, (**b**) contact load of 800 N.

Figure 7 shows the infrared thermal image of the friction pairs under different sliding distances. It clearly shows the variation characteristics of the surface temperature during the friction test. The color of the wire rope changes greatly with the increasing sliding distance, especially in the early stage of the test. When the sliding distance increases from 5 m to 50 m, the surface color of the wire rope gradually changes from dull to bright, as shown in Figure 7a–d. This means that the surface temperature of the wire rope rises rapidly at this stage. Additionally, the frictional heat generated by the test is gradually diffused from the contact surface, and the temperature of the pulley groove rises significantly faster (see Figure 7c). This is because the structure of the groove is regular and the surface is complete and continuous. The friction heat transfers faster on the pulley. Furthermore, when the sliding distance exceeds 100 m, the surface color of the wire rope is very bright and basically unchanged. However, the temperature of the contact area is still increasing with the sliding distance, as shown in Figure 7e–h. Therefore, the friction heat generated by the rope–pulley friction transfers slowly on the surface of the wire rope, and the temperature rise is mainly concentrated in the sliding contact area. The temperature variation in the wire rope is more easily observed by the infrared thermal imager at the beginning of the test.

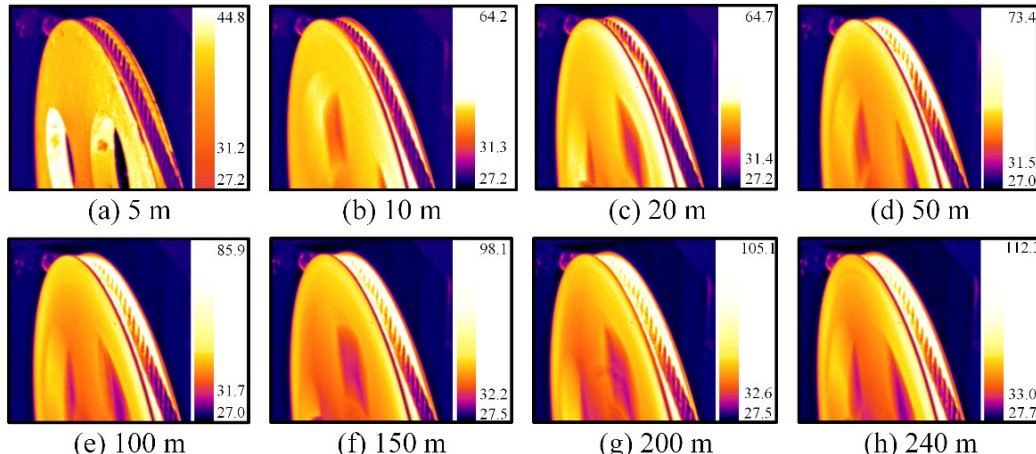

(a) 5 m
(b) 10 m
(c) 20 m
(d) 50 m
(e) 100 m
(f) 150 m
(g) 200 m
(h) 240 m

**Figure 7.** Infrared thermal images of wire rope under different sliding distances.

The variation in the maximum temperature rise of the wire rope with the sliding distance and the contact load collected by the infrared thermal imager is shown in Figure 8. The difference among the temperature rise curves under different contact loads is small, as shown in Figure 8a. The changing trend of the curves is basically the same. The curves increase rapidly at the beginning of the test but fluctuate greatly. This is because the structure of the rope surface changes greatly in the initial wear stage. The friction pair changes from

discontinuous line contact state to more stable surface contact state. Therefore, the surface temperature of the wire rope changes greatly in a short sliding distance. Additionally, the contact position of the wire rope on the rope groove changes greatly at this stage, and the surface temperature distribution of the wire rope is not uniform (see Figure 7). The location of highest surface temperature monitored by infrared thermography is constantly changing. Thus, the maximum temperature rise curves of the rope surface fluctuates greatly within a sliding distance of approximately 50 m. Furthermore, as the sliding distance continues to increase (from 50 m to 240 m), the temperature curves gradually stabilize after a slow growth stage. This indicates that the sliding wear surface of the wire rope gradually becomes stable. Although friction heat continues to be generated during the test, the surface temperature rise of the wire rope will not always increase. When the sliding distance exceeds 150 m, the heat generated by the rope–pulley sliding friction is balanced with the heat dissipation conditions of the surrounding environment. Thus, the maximum temperature rise of the wire rope is almost constant at the later stage of the friction test. Moreover, when the contact loads are 750 N and 800 N, the temperature rise curves rise significantly faster than other curves in the slow growth stage (from approximately 50 m to 150 m). This means that the increase in contact load will accelerate the increase in surface temperature of the wire rope during the friction test. This is because the larger contact load leads to an increase in the surface stress of the wire rope, and the evolution speed of the wear will be accelerated during the sliding process. Especially in the transition phase, the wear process will be more severe. Additionally, large contact stress will cause plastic deformation of the rope surface and the wear surface will be relatively smooth. Thus, more energy is converted into frictional heat during sliding friction.

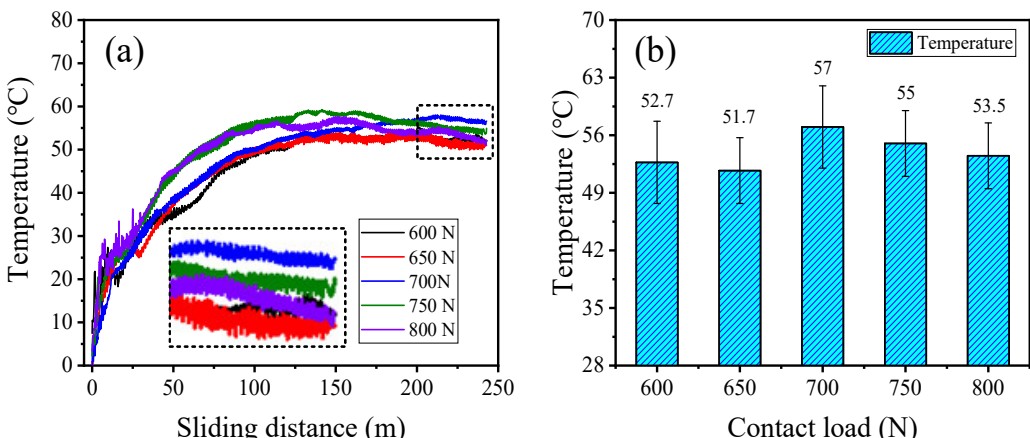

**Figure 8.** Variation curves of friction temperature rise under different contact loads: (**a**) process variation curves with sliding distance, (**b**) average temperature at the end of the measurement.

To quantitatively compare the influence of different contact loads on the friction temperature rise in the relatively stable stage, the average temperature rise of the rope surface was calculated based on the data collected from the last 50 m sliding distance, as shown in Figure 8b. The influence of contact load on the temperature rise is not obvious, and the change law is opposite to that of the COF. With the contact load increasing from 600 N to 800 N, the friction temperature rise fluctuates in the range of approximately 51 °C to approximately 57 °C. This indicates that the surface wear state of the wire ropes is similar in the relatively stable stage. Because the contact load has a greater influence on the progress and degree of surface wear, the influence on the surface characteristics and friction contact state is relatively small. Furthermore, when the contact load is 700 N, the friction temperature rise is maximized. This is because the frictional heat generated by the wire rope during sliding is closely related to the change in friction and wear morphology [37]. Additionally, the test temperature affects the hardness and tribological properties of metal materials [38]. Under the condition of small COF, the surface of the wire rope is relatively

smooth and the wear rate slows down. At this time, more energy is converted into friction heat during the sliding friction process. Therefore, the friction temperature rise of the wire rope is relatively large. Therefore, by monitoring the temperature change in the wire rope, the wear state can be well judged, but it is difficult to distinguish the change in contact load between the rope and the pulley.

*3.2. Characteristic Parameters of Surface Wear Scar*

The wear scar characteristics of the wire rope under different contact loads are presented in Figure 9. The wear surface of the wire rope is divided into many independent areas by the rope strand. This is because, under the condition of wire rope–pulley sliding contact, the rope will have multiple strands in contact with the groove at the same time, resulting in surface wear of multiple strands. Additionally, the contour of the wear scar on the strand surface is very irregular. This is due to the strand being made of multiple steel wires, which are cylindrical in structure. Thus, the wear edge of each wire is a circular arc, resulting in the wear scar on the whole rope strand being elliptical with irregular contours. Furthermore, the size of the wear scar tends to expand, as shown in Figure 9c. The wear scar contour map is measured and extracted by an optical microscope. It can be used to more clearly compare changes in the wear area of the wire rope surface. This is because the steel wire and the rope strand inside the wire rope will deform and move relatively under the action of external force [39]. Thus, the larger contact load will flatten the wire rope and cause structural deformation. Moreover, the larger contact load results in an increase in the surface stress of the steel wire, and the material is more prone to plastic deformation and shear spalling during sliding contact. Therefore, contact load affects the wear rate of the wire rope. In engineering application, the contact load between the rope and pulley should be minimized to protect the crane wire rope.

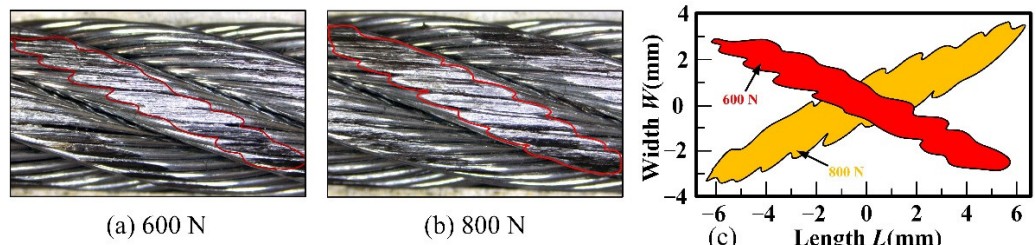

| (a) 600 N | (b) 800 N | (c) |

**Figure 9.** Distribution of wear scars on rope strand under different contact loads: (**a**,**b**) wear scars, (**c**) profile of wear scar under different contact loads.

Figure 10 shows the maximum wear width of the wear scar on the rope surface and the wear loss of the friction pairs under different contact loads. Wear width increases significantly with increasing contact load, from approximately 1.94 mm to 2.45 mm. Additionally, when the contact load is small (less than 700 N), the wear width increases slowly. The variation range in the wear width is approximately 0.14 mm. However, when the contact load increases from 700 N to 800 N, the wear width increases rapidly from approximately 2.08 mm to 2.45 mm. This indicates that the wear width increases nonlinearly with increasing contact load. This is not only because the surface wear is more severe but also the result of structural deformation of the wire rope under transverse pressure. The wear loss between the wire rope and the pulley under different contact loads is presented in Figure 9b. It increases linearly with the contact load (from approximately 5.614 g to 6.581 g). There are obvious differences in the change trend between the wear width and the wear loss. This is because the pulley groove during the sliding friction test also resulted in severe wear and produced a large amount of wear debris. Because the wear of the rope groove is relatively uniform during sliding wear, the amount of wear generated by the friction pairs increases steadily as the contact load increases. Therefore, by quantitatively analyzing the wear characteristics, it can be found that the actual damage of the contact load to the wear degree of the wire rope is approximately linear.

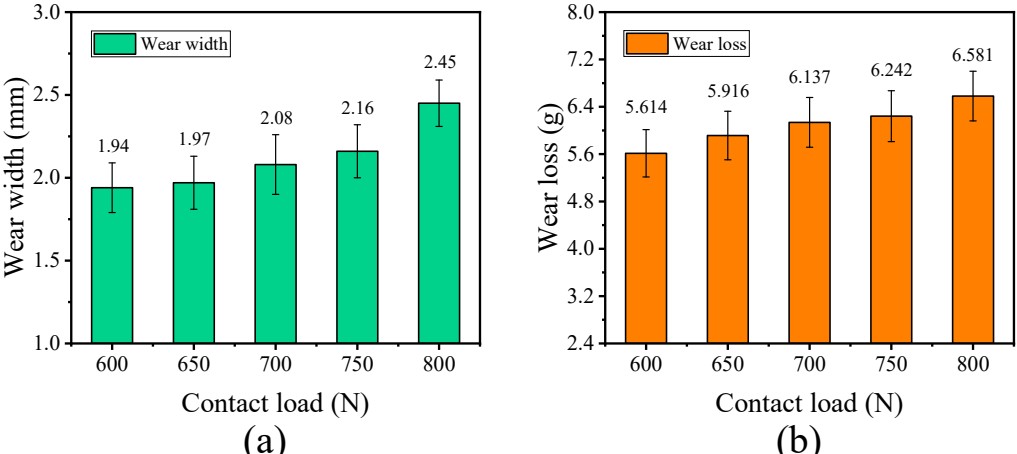

**Figure 10.** Wear characteristic parameters of wire rope under different contact loads: (**a**) maximum wear width, (**b**) wear loss.

### 3.3. Surface Wear Mechanism of Wire Rope

Figure 11 shows the optical microscopy of the wear surface of the rope strand under different contact loads. The difference in surface characteristics among the wear scars is obvious. When the contact load increases from 600 N to 800 N, the wear surface becomes smoother. When the contact load is small (see Figure 11a,b), the wear surface on the wires is rough and there are obvious scratches along the sliding direction. This indicates that a serious ploughing action occurs during the sliding friction between the wire rope and the pulley groove. Additionally, large wear particles in the gap between the wires can be observed (see Figure 11b). This means that there is a large amount of wear debris between the sliding friction pairs and the wear surface is scratched during the test. Furthermore, as the contact load continues to increase, the characteristics of the wear scar surface become clearer. When the load is 700 N (see Figure 11c), the steel wire surface material is squeezed seriously. Part of the material is misaligned and fills the gap between the wires, resulting in a continuous wear surface. This indicates that the main material removal forms of the wire rope change during the sliding wear process. The ploughing begins to weaken, and then the plastic deformation and spalling become more pronounced. As shown in Figure 11d, the surface is smooth and the furrows are difficult to distinguish. This is because the larger contact load makes the surface asperities in the contact area squeezed and deformed. During the sliding process, it will be directly rolled and flatten rather than ploughed. Therefore, the wear surface becomes smoother. Meanwhile, the wear debris in the middle of the sliding friction pairs is also crushed, further weakening the ploughing effect of the material. Moreover, with the contact load increasing to 800 N (see Figure 11e), the wear surface is smoother and the plastic deformation of the steel wire material is more obvious. Additionally, there are significant pitting and high-temperature oxidation characteristics caused by frictional heat on wear surfaces [40]. This is because the temperature of the friction surface of the wire rope increases significantly under large contact load. This causes the material in the contact region to soften, which in turn is more likely to be cut and plastically deformed. The contact surface of the wire rope is more likely to produce material shear and plastic deformation along the sliding direction [41]. Under the combined effect of transient high temperature and friction, the surface of wire rope is oxidized and discolored. Therefore, in the rope–pulley sliding friction and wear system, the variation in the contact load will cause the change in wear surface characteristics. The greater the contact load, the smoother the wear area in which contact stress and friction heat are the key factors affecting the wear characteristics of the crane wire rope.

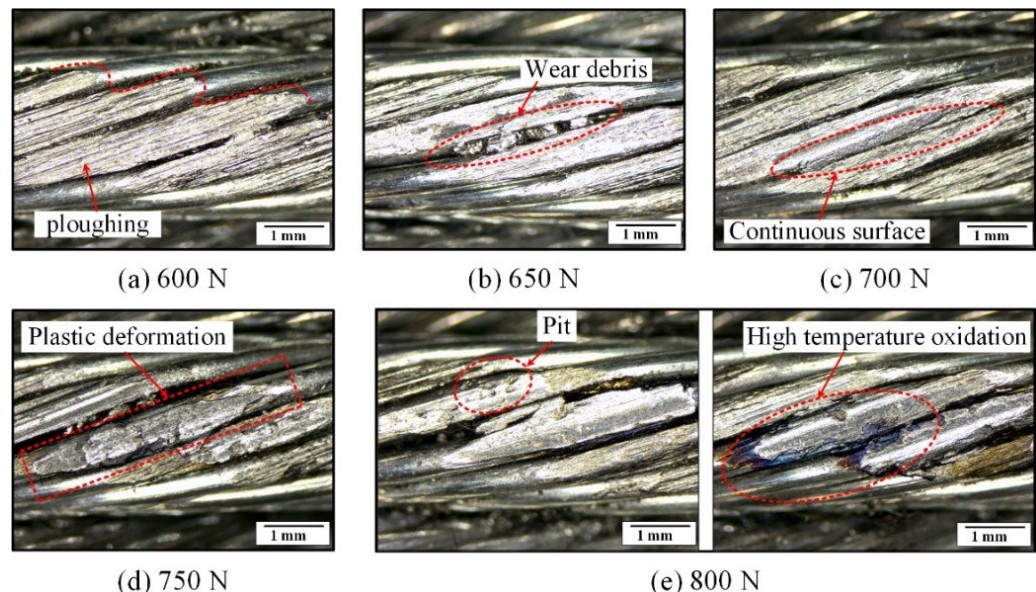

**Figure 11.** Surface wear characteristics of wire rope under different contact loads: (**a**) 600 N, (**b**) 650 N, (**c**) 700 N, (**d**) 750 N, (**e**) 800 N.

The morphological characteristics of the wear debris were obtained by the optical microscope and SEM, as shown in Figure 12. Figure 12a–c shows the optical micrographs of the wear debris in different sizes, and Figure 12d–f shows the SEM images of the wear debris in different sizes. They are taken from different test samples. It is observed that the characteristics of the wear debris produced during the friction test are constantly changing. The particle size of the wear debris is relatively large at the initial stage of the test, as shown in Figure 12a,d. The wear debris is mostly long strip, and there is obvious plastic deformation and furrows on the surface. This is because the contact area between the wire rope and the rope groove is small and the contact stress is large in the early stage of the friction test. The line contact state is easy to produce ploughing on the surface of the rope groove, resulting in removal of large particle materials. Furthermore, with the sliding distance increasing, the contact area between the friction pairs expands rapidly and the size of the wear debris becomes smaller, showing irregular flake particles (see Figure 12b,e). This indicates that the ploughing action of the wire rope on the surface of the groove is weakened, and the removal of the material is mainly caused by adhesion and spalling between the contact surfaces. Moreover, when the friction experiment enters a relatively stable stage, the wear rate of the friction pair slows down and the size of the wear debris becomes smaller (see Figure 12c,f). This is because the surface of the friction pairs becomes more and more smooth, and the material does not easily fall off during the sliding friction process. Additionally, part of the wear debris is crushed between the sliding contact surfaces and the wear particles become smaller and thinner. Therefore, in the process of sliding wear between the wire rope and pulley, the size of wear debris becomes smaller and smaller. In the relatively stable stage, the wear debris is the smallest and plays a role of lubrication in the sliding process, which can reduce the degree of the surface wear.

Figure 13 shows the SEM image of the wear surface on the damaged steel wires under different contact loads. This clearly shows the change in wear characteristics on rope surface with increasing contact load. The wear characteristic of the damaged wire is mainly furrows when the contact load is 600 N (see Figure 13a). The surface is uneven, and there are characteristics of plastic deformation caused by large contact stress. This means that the surface damage of the wire rope is obviously affected by its structure, and removal of the material is mainly caused by the mechanical damage. Because the wire rope is a spiral structure, the sliding contact surface is composed of discontinuous steel wires and rope strands. This causes the friction pair contact surface to be very uneven. Thus, the

sliding resistance is larger and the surface scratches and ploughing effect is more obvious. Additionally, the wear particles between the contact surface will exacerbate the surface ploughing. Furthermore, as the contact load changes to 650 N (see Figure 13b), although many furrows are distributed along the sliding direction on the wear surface, the size of the furrows is small and the surface is more regular. This indicates that an increase in the contact load will reduce the size of the furrows and make the spalling and pitting characteristics more pronounced. Moreover, when the contact load exceeds 700 N, the wear surface of the steel wire becomes very smooth (see Figure 13c–e). Especially when the contact load is 800 N, it is difficult to see the obvious damage characteristics in the wear area. This is because the friction temperature rise of the wire rope is larger under the condition of large contact load. The surface material will soften during the sliding wear process, which is more likely to be sheared and plastically deformed. Therefore, the furrow characteristics of the wear surface are weakened, and the adhesion characteristics, such as pitting and spalling, are enhanced. Therefore, with the increases in contact load, the furrows and spalling pits on the wear surface of the wire rope are significantly reduced. The abrasive wear is reduced during this process. Additionally, as the surface becomes smoother, the wear rate slows down, and adhesion is enhanced and the size of the wear debris becomes smaller. This also causes the abrasive wear characteristics to weaken. The sliding wear mechanisms between the wire rope and the pulley are mainly abrasive wear and adhesive wear. Furthermore, the morphology characteristics of abrasive wear on the wire rope surface are more obvious under smaller contact load. The effect of contact load on the wear mechanism is mainly achieved by plastic deformation and friction temperature rise.

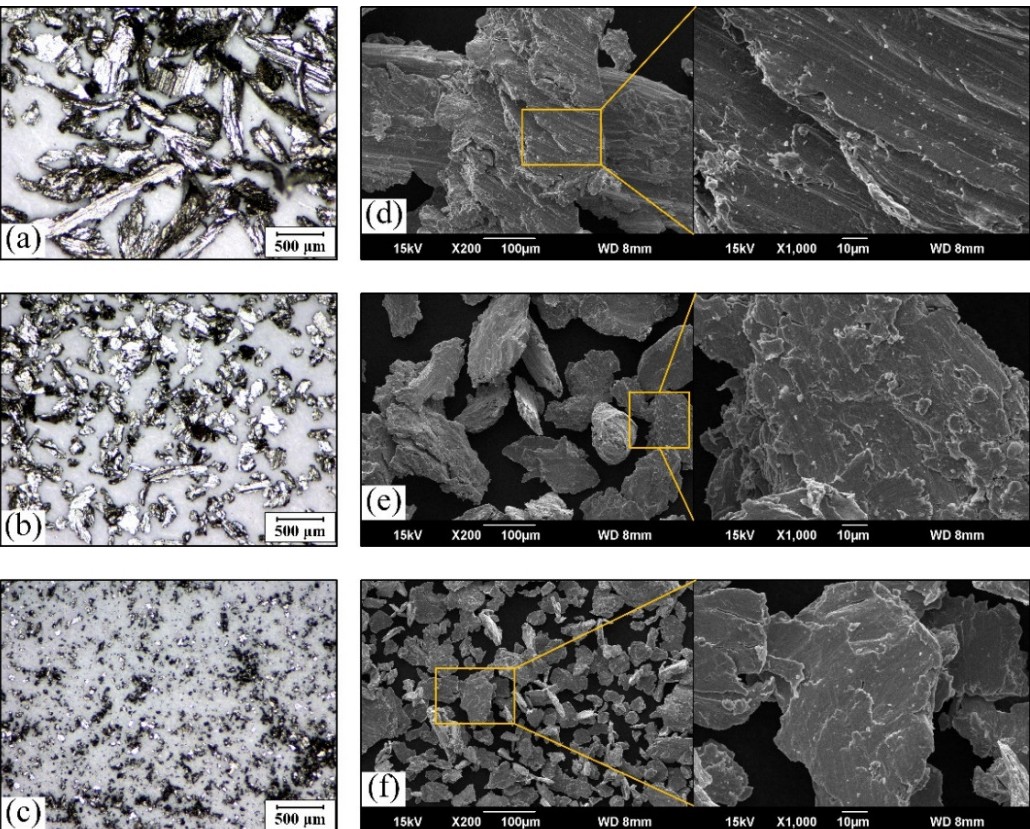

**Figure 12.** Optical microscopy and SEM of wear debris between wire rope and pulley: (**a**–**c**) macrograph under optical microscope, (**d**–**f**) microtopography under SEM.

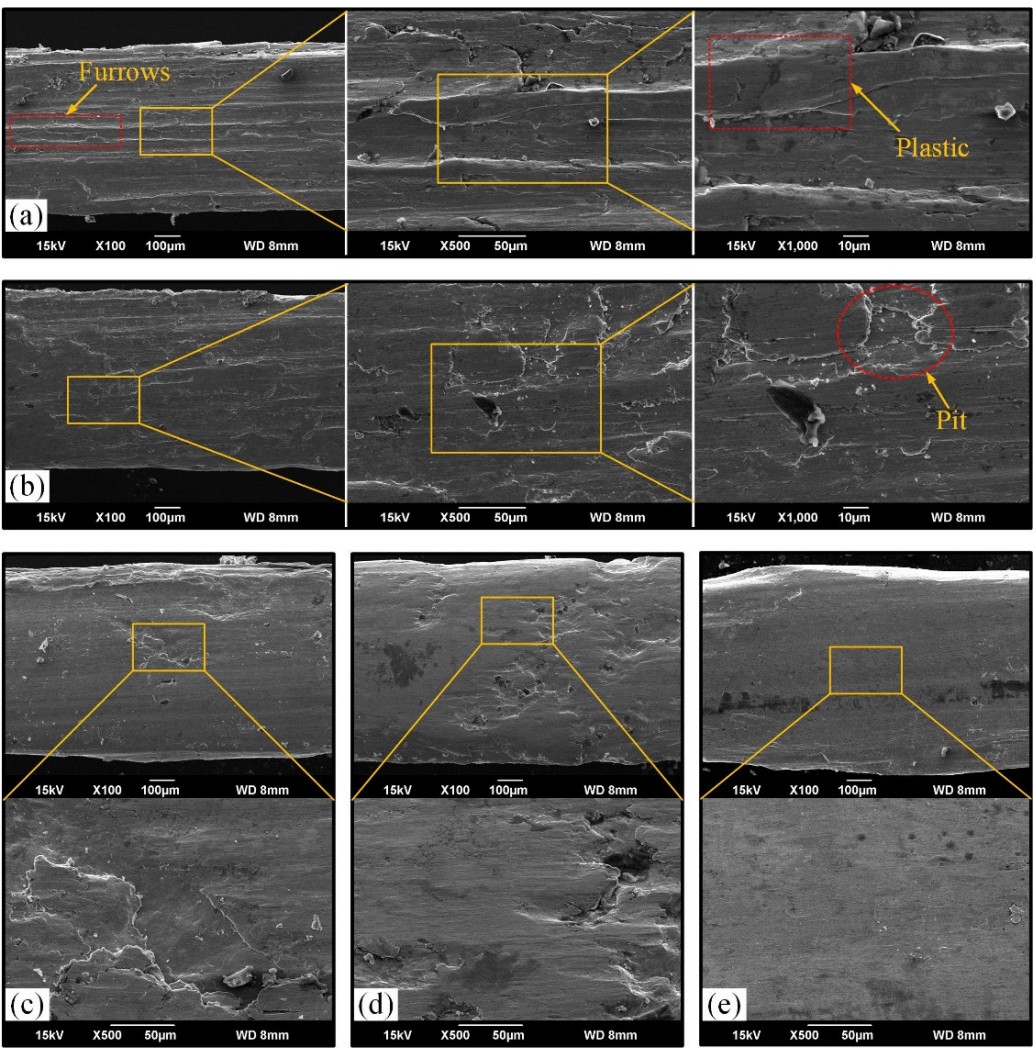

**Figure 13.** SEM image of worn steel wires under different contact loads: (**a**) 600 N, (**b**) 650 N, (**c**) 700 N, (**d**) 750 N, (**e**) 800 N.

### 3.4. Bending Fatigue Failure of Worn Wire Rope

The corresponding relationship between number of broken wires and bending cycles is obtained by carrying out bending fatigue tests on the wire rope with different surface wear, as shown in Figure 14. With the accumulation of bending fatigue time, the number of broken wires increases continuously. When the wire rope is not subjected to surface wear, the first broken wire occurs after approximately 5200 bending fatigue cycles, as shown in Figure 14a. Then, as the amount of bending increases, the broken wire increases linearly. However, when the number of broken wires exceeds four, its growth rate begins to slow down. Furthermore, when the number of broken wires exceeds six, its growth rate begins to increase again. With the increase in bending fatigue time, the broken wire curves rise approximately in a straight line. This indicates that the bending fatigue damage to the wire rope is a cumulative deterioration process, and, with an increase in bending fatigue time, the fracture speed of the wires is accelerated. Additionally, the variation trend of the broken wire curves of the rope samples is basically the same, but the anti-fatigue performance is obviously degraded. With an increase in contact load, the bending fatigue times of the wire rope under the same broken wire condition decrease. The first broken wire begins to appear on these rope samples after 2000–3500 bending fatigue cycles. This means that surface wear has a significant effect on the bending fatigue strength of wire ropes and accelerates the fracture speed of internal steel wires [42]. Furthermore, by comparing the change process of the curves, it can be seen that, when the number of broken wires exceeds

approximately seven, the broken wires change more regularly with increasing bending fatigue times. This indicates that the internal wire breaking of the wire rope has a certain randomness in the early stage of the bending fatigue test, and the breaking sequence of different positions is uncertain. However, when the broken wire reaches a certain number, the bending fatigue dangerous area of the wire rope is basically determined, and the stress becomes concentrated near the broken wire of the wire rope. This is more likely to cause a fracture of the surrounding steel wires. Therefore, the rope samples in the bending fatigue process appear as continuous concentrated broken wires in the late bending fatigue test. Moreover, Figure 14b shows the maximum bending fatigue times of the rope sample under different contact loads. The bending fatigue times decrease from approximately 7700 to 4850 with increasing contact load. The bending fatigue life of the wire rope is obviously degraded, and it is proved that the harm of sliding wear to the wire rope increases with increasing contact load. Therefore, controlling the contact load between the crane wire rope and the pulley can effectively extend its bending fatigue life.

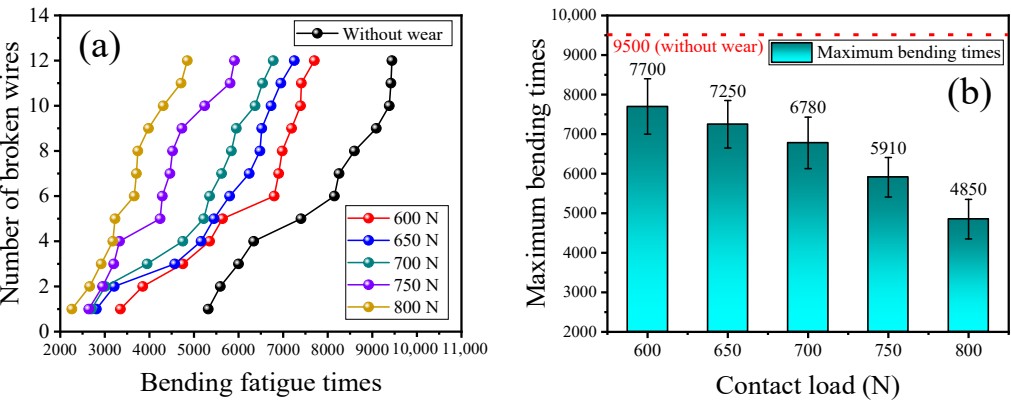

**Figure 14.** Bending fatigue degradation characteristics of different rope samples: (**a**) variation curves of broken wire number with bending fatigue times, (**b**) maximum bending fatigue times under different contact loads.

Figure 15 shows the broken wire distribution (Figure 15a–c) and fracture morphology (Figure 15c–e) of rope samples caused by bending fatigue test. The observations in Figure 15 are taken from different rope samples and wire samples. The broken wires produced during the bending fatigue process are mainly concentrated in the wear region. This indicates that surface wear reduces the bending fatigue strength of wire rope [42]. Additionally, the fracture of the steel wire appears in the middle of the wear scar and is distributed along the spiral direction on the rope strand (see Figure 15a–c). This is because wire tension is along the steel wire axial direction in the process of bending and the fracture is along the steel wire radial direction [43]. Therefore, the broken steel wire is continuously distributed along the strand axis. Additionally, the fracture of the steel wire is very neat, indicating that the fracture is a transient process, and no plastic deformation caused by a large load is produced before fracture. Figure 15d shows the fracture characteristics of the unworn steel wire. The fracture profile of the steel wire is very complete and there are many cracks on the fracture surface. This indicates that the internal damage of steel wire under bending fatigue load is a process of continuous cumulative change. The crack propagates with the increase in bending fatigue times until fracture. Furthermore, when the steel wire is subjected to surface wear, the fracture characteristics do not change significantly, but the number of cracks on the surface increase and the size becomes larger. This means that surface wear accelerates growth in fatigue cracks, causing the wire rope to break faster. Moreover, the fracture morphology of the wire rope is relatively regular and the surface is bright crystalline. Therefore, the fracture mechanism of wire rope under bending fatigue load is mainly brittle fracture. Surface wear causes the broken wire to concentrate and speed up, greatly affecting the service life of the wire rope.

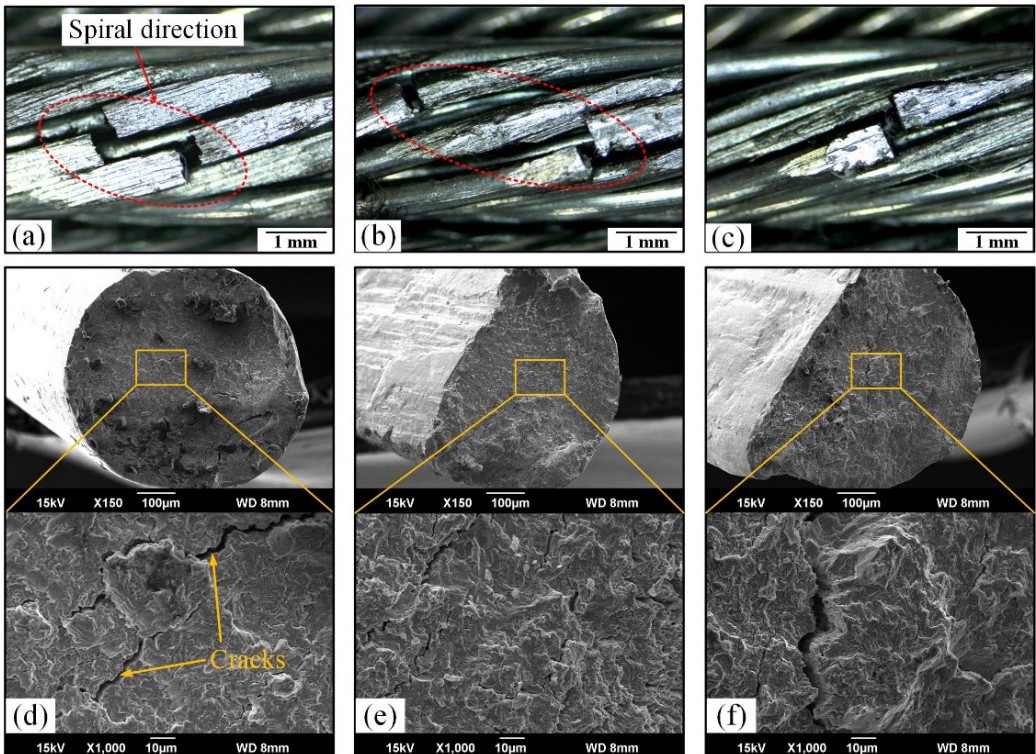

**Figure 15.** Fracture failure behavior of worn wire rope caused by bending fatigue: (**a**–**c**) distribution characteristics of broken wires, (**d**–**f**) fracture characteristics.

## 4. Conclusions

In this study, the friction characteristics and wear mechanisms between crane wire rope and a pulley groove were investigated. Then, the bending fatigue life and fracture failure behavior of the damaged rope samples were analyzed. The major conclusions are as follows:

(1) The COF decreases with the increasing sliding distance, and the variation speed is fast and then slow. The effect of contact load on the COF in the stable stage is small. With the contact load increasing from 600 N to 800 N, the COF fluctuates in the range of approximately 0.52 to 0.59.

(2) The surface temperature rise of the wire rope increases rapidly and then becomes stable gradually during the friction test. The surface temperature of the wire rope rises relatively fast under the condition of large load contact. The increase in contact load causes the friction temperature rise to increase first and then decrease, and the temperature reaches the maximum at the load of 700 N, which is approximately 57 °C.

(3) Rope–pulley sliding contact causes the wear of the wire rope to be discontinuously distributed on multiple strands with irregular contour. Under the same sliding condition, the larger contact load leads to an increase in the wear degree of the wire rope. The wear width increases from approximately 1.94 mm to 2.45 mm with the increasing contact load.

(4) In the process of sliding wear between the wire rope and the pulley groove, the size of wear debris becomes smaller and smaller. The wear surface of the wire rope becomes smoother and the damage characteristics, such as furrows and pits, are no longer obvious with the increasing contact load. The wear mechanism under large contact load is mainly adhesive wear.

(5) Surface wear accelerates the fracture speed of the wire rope and causes the broken wire position to be concentrated under bending fatigue condition. An increase in contact load leads the maximum bending fatigue cycles of the worn wire rope to decrease from approximately 7700 to 4850. The bending fatigue life of the wire rope

decreases with an increase in sliding contact load. The fracture failure mechanism of the wire rope is mainly brittle fracture.

**Author Contributions:** Conceptualization, X.C. (Xiangdong Chang) and X.C. (Xiao Chen); methodology, X.C. (Xiangdong Chang) and X.C. (Xiao Chen); validation, Y.D. and H.L.; investigation, Y.D. and W.T.; data curation, Q.Z. and K.H.; writing—original draft preparation, X.C. (Xiangdong Chang) and X.C. (Xiao Chen); writing—review and editing, Y.D. and H.L.; visualization, W.T., Q.Z. and K.H.; supervision, Y.D. and H.L.; project administration, X.C. (Xiangdong Chang); funding acquisition, X.C. (Xiangdong Chang). All authors have read and agreed to the published version of the manuscript.

**Funding:** This research was funded by National Natural Science Foundation of China, grant numbers 52005272 and 51975572. This project is also partly supported by Priority Academic Program Development of Jiangsu Higher Education Institutions (PAPD), Top-notch Academic Programs Project of Jiangsu Higher Education Institutions (TAPP).

**Data Availability Statement:** The data presented in this study are available on request from the corresponding author.

**Conflicts of Interest:** The authors declare no conflict of interest.

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
