# Peer review of "Friction and Wear Behavior between Crane Wire Rope and Pulley under Different Contact Loads"

_lubricants, doi:10.3390/lubricants10120337_

Round 1

Reviewer 1 Report

Regarding the article entitled: Friction and Wear Behavior between Crane Wire Rope and Pulley under Different Contact Loads, I have the following comments and suggestions.

It is necessary to supplement the results of the following studies:

1.      line 59: Cruzado et al. [11-13] .... what was the result of their investigation. (you can't just set research directions without reporting their results).

2.      line 75: Peng et al. [18,19] .... what was the result of their investigation. (you can't just set the direction of research without reporting their results).

3.      line 76: Oksanen et al. [20,21] .... what was the result of their investigation. (you can't just set the direction of research without reporting their results).

4.      line 80: Chang et al. [22] .... what was the result of their investigation. (you can't just set the direction of research without reporting their results).

5.      line 81: Jikal et al. [23] .... what was the result of their investigation. (you can't just set the direction of research without reporting their results).

6.      line 85: JWahid et al. [25] .... what was the result of their investigation. (you can't just set the direction of research without reporting their results).

7.      line 87: Battini et al. [26] .... what was the result of their investigation. (you can't just set the direction of research without reporting their results).

Materials and Methods:

1.      Please put the chemical composition of the material of the pulley and rope in the overview table and not in the text. and also insert the chemical range of individual alloying elements of both materials according to the standard.

2.      line 122: table 1. replace columns with rows. insert individual values into columns

Test procedure:

1.      figure 3: add text to the description of figure 3b (not and).

2.      What is the angle of the rope in Figure 3a? Add to the text of the given chapter.

3.      During the wear test process, is the rope tightly clamped at both ends or does it wind up? Write in the text of the given chapter.

4.      figure 3: consider adding the description to figure 3e and 3f.

2.3.  Test parameters and methods:

1.      line 167: Figure 4a and d (should be labeled 4c and d).

2.      insert zoom scales in Figures 4c and 4f.

3.      figure 4a: delete

4.      figure 4b: change the text color from blue to e.g. yellow (the blue color of the text blends with the image)

5.      figure 4d: an explanation of the color scale is missing (what is the value of e.g. red?).

6.      Consider inserting the description of Figure 4 into the description of Figure 4. (a).....(b).....etc.

3.1. Sliding friction parameters:

1. What is the reason for the drop in COF at a load of 700N in Fig. 5?

2. figure 6a: delete (still repeated in the article).

3.      figures 6b,c,d,e,f: all are graphically the same. leave only figures 6b and 6f, which represent extreme values of temperatures.

4.      Shorten the text for the description of Figure 6, line 212-240.

5.      figure 8a-b: leave only Temperature [°C] on the y-axes

6.      figure 8b: change the legend from Temperature rise to the average temperature at the end of the measurement.

7.      For what reason was the highest temperature 57 °C with a load of 700 N. The connection between this temperature and the lowest COF value should be described.

Characteristic parameters of surface wear scar:

1.      figure 9: keep only figures 9a and 9e. only limit values for optical comparison.

2.      figure 9f: either edit because the overlap of layers is confusing, or leave only the values from figures 9a and 9e for example. or insert numerical wear values into a graph or table.

3.      Shorten the description text from line 309-329, leaving only the main result of this paragraph.

3.3.           Surface wear mechanism of wire rope:

1.      Please, in the text between lines 390-412, insert the text that describes the measurement from which the fragment samples were taken.

3.4.  Bending Fatigue failure of worn wire rope:

1.      In the text between lines 480-501, insert the text that describes the measurement parameters from where the individual ropes that were examined for fracture were taken. Insert this label also in the description of Figure 15.

4.      Conclusions:

1.      statement number 1: is half true, because the COF decreases, but from a load of 700 N and more, it already has an increasing character. (I propose to edit and describe the reason why such a result is achieved)

2.      statement number 2: again explain in more detail why the highest temperature was reached at a load of 700 N and not at the highest load of 850 °C.

3.      statement number 3: The wear width increases from approximately 1.94 mm to 2.45 mm with increasing contact load. (depending on which parameter - add)

4.      statement number 5: Surface wear accelerates the fracture speed of the wire rope and causes the broken wire position to be concentrated under bending fatigue condition. The increase of contact load leads the maximum bending fatigue cycles of the worn wire rope to decrease from approximately 7700 to 4850.

(again in view of which parameter - to add)?

Please answer the question:

In practice, the rope moves along the pulley, which performs a rotating movement. What is the practical reason for the static friction of the rope on the pulley?

Well thank you.

Adding information to the article.

1. Please add information or pictures - the microstructure of the rope + its final heat treatment.

2. In the experimental part of the article, I did not find a single comparison with the results of other authors. Fill in please.

3. Some text is unnecessary in the experimental part, you do not need to re-describe the description of individual measurements at the beginning of each section, go straight to the results.

The scope of the article is a bit long in terms of the number of pages, so it is necessary to add references to at least 35-40 authors.

Possible addition of literary sources from MDPI publishing house:

Studeny, Z.; Krbata, M.; Dobrocky, D.; Eckert, M.; Ciger, R.; Kohutiar, M.; Mikus, P. Analysis of Tribological Properties of Powdered Tool Steels M390 and M398 in Contact with Al2O3. Materials 2022, 15, 7562. https://doi.org/10.3390/ma15217562

I ask the authors of the article to highlight each corrected or supplemented text in the article in yellow.

well thank you

Author Response

Thank you for your comments. We have studied the comments carefully and have made revisions and modifications which we hope meet with approval. Additionally, the revised portions are marked up using “Track Changes” in revised manuscript. The main corrections in the paper and the responds to the comments are as follows:

It is necessary to supplement the results of the following studies:

Response: Thank you for your advice. We have supplemented the research results of the relevant literature in the manuscript and marked in yellow.

  1. line 59: Cruzado et al. [11-13] .... what was the result of their investigation. (you can't just set research directions without reporting their results).

Response: The statements “The results show that the wear volume caused by fretting between steel wires depends mainly on the variation of normal contact load.” were added. (Page 2, Line 61-62)

  1. line 75: Peng et al. [18,19] .... what was the result of their investigation. (you can't just set the direction of research without reporting their results).

Response: The statements “They found that effective lubrication reduces impact vibration and the low tempera-tures increase the COF between wire ropes.” were added. (Page 3, Line 77-78)

  1. line 76: Oksanen et al. [20,21] .... what was the result of their investigation. (you can't just set the direction of research without reporting their results).

Response: The statements “The results show that the material removal proceeds through the spalling of the de-formation tongues by crack growth between the graphite nodules and the surface.” were added. (Page 3, Line 81-83)

  1. line 80: Chang et al. [22] .... what was the result of their investigation. (you can't just set the direction of research without reporting their results).

Response: The statements “They found that the surface wear causes the breaking elongate of the wire rope to de-crease and changes the fracture morphology.” were added. (Page 3, Line 86-88)

  1. line 81: Jikal et al. [23] .... what was the result of their investigation. (you can't just set the direction of research without reporting their results).

Response: The statements “They found that the corrosion could lead to a significant decrease in fatigue strength and accelerate fatigue damage of corroded steel strands. Fatigue life decreases linearly with the corrosion level.” were added. (Page 3, Line 90-93)

  1. line 85: JWahid et al. [25] .... what was the result of their investigation. (you can't just set the direction of research without reporting their results).

Response: The statements “By analyzing the mechanical and chemical properties of wire rope, they found that the behavior of wire rope is semi-brittle.” were added. (Page 3, Line 96-98)

  1. line 87: Battini et al. [26] .... what was the result of their investigation. (you can't just set the direction of research without reporting their results).

Response: This part of the description is rewritten. the statements “Battini et al. [26] proposed a thermal method for estimating fatigue life of metallic ropes. The proposed model showed a very good correlation between early-stage data and first wire failure conditions. Then, the research results realized the evaluation of damage and life of wire ropes by monitoring heat changes.” were added. (Page 3, Line 98-101)

Test procedure:

  1. figure 3: add text to the description of figure 3b (not and).

Response: The description of Figure 3b “(b) front view of friction tester” was added. (Page 6, Line 189)

  1. What is the angle of the rope in Figure 3a? Add to the text of the given chapter.

Response: The statement for the contact angle “The contact angle of the wire rope is 45°.” was added. (Page 5, Line 158-159)

  1. During the wear test process, is the rope tightly clamped at both ends or does it wind up? Write in the text of the given chapter.

Response: The rope is tightly clamped at both ends. The statement “During the wear test, the wire rope is clamped firmly at both ends.” was added. (Page 5, Line 165-166)

  1. figure 3: consider adding the description to figure 3e and 3f.

Response: The description for Figure 3e and f “(e) Tensioning and fixing of wire rope, (f) bending contact state of wire rope.” was added. (Page 6, Line 190-191)

2.3. Test parameters and methods:

  1. line 167: Figure 4a and d (should be labeled 4c and d).

Response: We have revised this part. As we adjusted the order of the pictures, “Figure 4a and d” was changed to “Figure 4b and c”. (Page 7, Line 211)

  1. insert zoom scales in Figures 4c and 4f.

Response: The scales were added in Figure 4c and e in the revised Figure 4. (Page 7, Line 221)

  1. figure 4a: delete

Response: We have deleted it. (Page 7, Line 221)

  1. figure 4b: change the text color from blue to e.g. yellow (the blue color of the text blends with the image)

Response: The text color has been changed to yellow, as shown in Figure 4a in the revised Figure 4. (Page 7, Line 221)

  1. figure 4d: an explanation of the color scale is missing (what is the value of e.g. red?).

Response: The color in Figure 4b represents the relative heights of the wire rope. The description “The structural contour of the wire rope is distinguished by the color change in the 3D diagram, and the same color corresponds to the same height. The redder the color, the greater the relative height.” was added to explain the color scale of Figure 4b in the revised manuscript. (Page 7, Line 211-213)

  1. Consider inserting the description of Figure 4 into the description of Figure 4. (a).....(b).....etc.

Response: The description for Figure 4a-e “(a) infrared thermal image, (b) 3D topography of wear scar, (c) optical micrograph of wear scar, (d) SEM of wear surface, (e) fatigue broken wires.” was added. (Page 7, Line 222-224)

3.1. Sliding friction parameters:

  1. What is the reason for the drop in COF at a load of 700N in Fig. 5?

Response: Thank you for your comment, we added the explanation of the test results. In the process of friction experiment, the change of load mainly affects the contact state and surface wear characteristics of the wire rope. The increase of the load leads to the increase of the effective contact area of the friction pair, resulting in the increase of the COF. At the same time, the increase of load will strengthen the plastic deformation of metal material surface and make the sliding contact surface smoother. This will reduce the COF. Therefore, in the process of load increase, the two variation characteristics of COF are balanced each other, and the extreme value appears when the load reaches 700 N. Moreover, the statements “The variation of the COF is due to the change of the surface characteristics of the wire rope under different contact loads. Additionally, the structure of wire rope is complex, and the change of its contact state during the experiment will also have a great influence on the COF [34]. The increase of contact load will increase the effective contact area of metal friction pair, resulting in the increase of the COF. However, during the sliding process, the larger load is more likely to cause plastic deformation of the micro-protrusions on the contact surface, which in turn causes the surface pits and protrusions to complement each other. This ultimately results in a smoother sliding contact surface of the wire rope. The COF under this sliding condition is relatively smaller. Therefore, in the process of increasing the contact load, the two COF variation characteristics are balanced with each other. The COF will have a minimum value. This shows that the smooth surface of the wire rope plays a leading role in the COF when the contact load is 700 N. Furthermore, as the contact load continues to increase, the adhesion force increases, and the COF increases during the sliding process.” were added in the revised manuscript. (Page 8, Line 258-271)

  1. figure 6a: delete (still repeated in the article).

Response: Thank you for your advice. We have deleted it. (Page 9, Line 307)

  1. figures 6b,c,d,e,f: all are graphically the same. leave only figures 6b and 6f, which represent extreme values of temperatures.

Response: Thank you for your advice. We have deleted Figure 6c, d and e. Only Figure 6b and f are left. (Page 9, Line 307)

  1. Shorten the text for the description of Figure 6, line 212-240.

Response: Thank you for your advice. After modifying Figure 6, we deleted the relevant descriptions and only retained important conclusions and explanations. (Page 8-9, Line 279-306)

  1. figure 8a-b: leave only Temperature [°C] on the y-axes

Response: Thank you for your advice. We modified the y-axis. It has been changed to “Temperature (°C)”, as shown in Figure 8. (Page 11, Line 398)

  1. figure 8b: change the legend from Temperature rise to the average temperature at the end of the measurement.

Response: Thank you for your advice. It has been changed to “(b) average temperature at the end of the measurement”. (Page 11, Line 400)

  1. For what reason was the highest temperature 57 °C with a load of 700 N. The connection between this temperature and the lowest COF value should be described.

Response: Thank you for your advice. In order to explain the test results of temperature change, the statements “Furthermore, when the contact load is 700 N, the friction temperature rise is maximized. This is because the frictional heat generated by the wire rope during sliding is closely related to the change of friction and wear morphology [35]. Additionally, the test temperature affects the hardness and tribological properties of metal materials [36]. Under the condition of small COF, the surface of the wire rope is relatively smooth, and the wear rate slows down. At this time, more energy is converted into friction heat during the sliding friction process. Therefore, the friction temperature rise of the wire rope is relatively large.” were added in the revised manuscript. (Page 11, Line 387-395)

Characteristic parameters of surface wear scar:

  1. figure 9: keep only figures 9a and 9e. only limit values for optical comparison.

Response: Thank you for your advice. We have modified Figure 9. Figure 9b, c and d were deleted. (Page 12, Line 430)

  1. figure 9f: either edit because the overlap of layers is confusing, or leave only the values from figures 9a and 9e for example. or insert numerical wear values into a graph or table.

Response: Thank you for your advice. We only keep the contour plots of Figure 9a and e for comparison. The revised picture is presented in Figure 9c, in the revised manuscript. (Page 12, Line 430)

  1. Shorten the description text from line 309-329, leaving only the main result of this paragraph.

Response: Thank you for your advice. Based on the modified Figure 9, we have deleted the relevant description and analysis. (Page 11, Line 402-420)

3.3. Surface wear mechanism of wire rope:

  1. Please, in the text between lines 390-412, insert the text that describes the measurement from which the fragment samples were taken.

Response: Thank you for your comment. We have added the relevant describes in the revised manuscript. The statements “Figure 12a-c show the optical micrographs of the wear debris in different sizes, and Figure 12d-f show the SEM images of the wear debris in different sizes. They are taken from different test samples.” were added. (Page 13, Line 498-500)

3.4. Bending Fatigue failure of worn wire rope:

  1. In the text between lines 480-501, insert the text that describes the measurement parameters from where the individual ropes that were examined for fracture were taken. Insert this label also in the description of Figure 15.

Response: Thank you for your comment. The observations in Figure 15 are taken from different rope samples and wire samples. Figure 15a-c shows the broken wire distribution in rope stands, and they are optical microscopic. Furthermore, Figure 15d-f shows the fracture morphology of different wires with surface wear they are SEM images. Moreover, the relevant description was given in the revised manuscript. (Page 17, Line 603-605)

  1. Conclusions:
  2. statement number 1: is half true, because the COF decreases, but from a load of 700 N and more, it already has an increasing character. (I propose to edit and describe the reason why such a result is achieved)

Response: Thank you for your advice. We have added an explanation of the variation of the COF in the revised manuscript. (Page 8, Line 258-271)

  1. statement number 2: again explain in more detail why the highest temperature was reached at a load of 700 N and not at the highest load of 850 °C.

Response: Thank you for your advice. We have added an explanation of the variation of the temperature rise of wire rope in the revised manuscript. (Page 11, Line 387-395)

  1. statement number 3: The wear width increases from approximately 1.94 mm to 2.45 mm with increasing contact load. (depending on which parameter - add)

Response: Thank you for your comments. This conclusion is derived from the analysis in Figure 10. (Page 12, Line 438)

  1. statement number 5: Surface wear accelerates the fracture speed of the wire rope and causes the broken wire position to be concentrated under bending fatigue condition. The increase of contact load leads the maximum bending fatigue cycles of the worn wire rope to decrease from approximately 7700 to 4850.

(again in view of which parameter - to add)?

Response: Thank you for your comments. This conclusion is derived from the analysis in Figure 14. (Page 16-17, Line 595)

Please answer the question:

In practice, the rope moves along the pulley, which performs a rotating movement. What is the practical reason for the static friction of the rope on the pulley?

Response: Although the rolling contact is the main contact form between the wire rope and the sheave during the operation of the transmission system, the friction and wear caused by the sliding contact is more serious to the wire rope. Moreover, the use frequency of the wire rope is high, and the wear is a process of continuous accumulation and deterioration during long-term service. Therefore, it is of great engineering significance to study the sliding friction characteristics between the wire rope and the sheave to en-sure the safe service of wire ropes and reduce the damage.

  1. Please add information or pictures - the microstructure of the rope + its final heat treatment.

Response: We have supplemented the steel wire processing technology and materials in the revised manuscript. The internal steel wires are manufactured by the cold drawing process from high quality carbon structural steel. The statement “The internal steel wires are manufactured by the cold drawing process from high quality carbon structural steel.” was added in the revised manuscript. (Page 4, Line 147-148)

  1. In the experimental part of the article, I did not find a single comparison with the results of other authors. Fill in please.

Response: Thank you for your advice. We have added references to support the analysis results, as follows:

1) Page 8, Line 261, reference [34].

2) Page 11, Line 390, reference [35].

3) Page 11, Line 391, reference [36].

4) Page 11, Line 414, reference [37].

5) Page 13, Line 484, reference [38].

6) Page 13, Line 488, reference [39].

7) Page 16, Line 583, reference [40].

8) Page 17, Line 608, reference [40].

9) Page 17, Line 611, reference [41].

  1. Some text is unnecessary in the experimental part, you do not need to re-describe the description of individual measurements at the beginning of each section, go straight to the results.

Response: Thank you for your advice. We have deleted these unnecessary descriptions in the manuscript.

The scope of the article is a bit long in terms of the number of pages, so it is necessary to add references to at least 35-40 authors.

Response: Thank you for your advice. We added 15 references to this paper and now have 41 references in the revised manuscript. (Page 3, Line 103-119) and (Page 20, Line 732-764)

Possible addition of literary sources from MDPI publishing house:

Studeny, Z.; Krbata, M.; Dobrocky, D.; Eckert, M.; Ciger, R.; Kohutiar, M.; Mikus, P. Analysis of Tribological Properties of Powdered Tool Steels M390 and M398 in Contact with Al2O3. Materials 2022, 15, 7562. https://doi.org/10.3390/ma15217562

Response: Thank you for your advice. We have cited this article in the revised manuscript. (Page 11, Line 391, reference [36])

I ask the authors of the article to highlight each corrected or supplemented text in the article in yellow.

Response: Thank you for your advice. All modifications in the text are marked yellow.

Reviewer 2 Report

The reviewed paper titled “Friction and Wear Behavior between Crane Wire Rope and pulley under Different Contact Loads” is of a good scientific level.  The  tribological behavior of the wire rope was demonstrated. Analyzing aims and scope of journal In my opinion, it is not quite suitable for publication in Lubricants, it lacks an analysis of eg the influence of lubricant on wear. Apart from this fact, I think that the reviewed work is very interesting and worth publishing.

 The authors did not avoid minor errors and inaccuracies. Notes for reflection and minor correction of the work:

 1. Introduction - the introduction is good, but you can add a section on methods to reduce rope wear and mention the effect of lubricants.

 2. Materials and Methods:

- it is worth specifying the diameter of the wires in the rope, were there different diameters of the wires in the rope?

- specify the mechanical properties of the rope and wires, it has an influence on the degree of rope wear

- no methodology for measuring temperature

 3. Results and discussion -  in general, this part of the work is relatively well described

- how was figure 9f made? no description what data?

 4. Conclusions

Autors said: The bending fatigue life of the wire rope decreases approximately linearly, there is not true; in some areas it may be close to linear and in others to logarithmic

Author Response

Thank you for your comments. We have studied the comments carefully and have made revisions and modifications which we hope meet with approval. Additionally, the revised portions are marked up using “Track Changes” in revised manuscript. The main corrections in the paper and the responds to the comments are as follows:

  1. Introduction - the introduction is good, but you can add a section on methods to reduce rope wear and mention the effect of lubricants.

Response: Thank you for your advice. We have added the section in Introduction. The statements “To improve the tribological behavior of the wire rope under different service conditions, Zhang et al. [27,28] studied the tribological behavior of wire rope under the condition of modified lubricating oil, and analyzed the effect of different additives on reducing the surface wear. They found that the lanthanum stearate modified lubricating oil can better reduce the wear of the wire rope under different sliding speeds and contact loads. Chang et al. [29] studied the tribological properties of lubricated wire ropes in different corrosive environments. The results show that the corrosion solutions degenerate the anti-friction and anti-wear properties of the lubricating grease and oil. Mc Coll et al. [30] studied the friction and wear properties of steel wire under different lubrication conditions, and found that grease lubrication can form a better protective layer on the contact surface and reduce the COF. Périer et al. [31] analyzed the fretting friction and wear behaviors of wires in NaCl solution and aqueous solution. The results show that the influence of NaCl solution on the fretting fatigue life is not obvious. Molnár et al. [32] studied the performance degradation of wire rope and internal wire in salt solution. It is helpful to predict the service life of wire ropes. Wu et al. [33] studied the influence of sulfide concentration, stress level and pH value on the stress corrosion cracking of steel wire. They found that galvanized coating was effective in reducing the corrosion.” were added to introduce the research progress of scholars on the lubrication of wire rope and wear reduction under different service conditions. (Page 3, Line 103-119)

  1. Materials and Methods:

1) It is worth specifying the diameter of the wires in the rope, were there different diameters of the wires in the rope?

Response: Thank you for your comments. The diameter of the internal steel wires is the same.

2) Specify the mechanical properties of the rope and wires, it has an influence on the degree of rope wear

Response: Thank you for your advice. The main mechanical properties of the wire rope were added in Table 1. (Page 4, Line 154)

3) No methodology for measuring temperature

Response: Thank you for your comment. The methodology for measuring temperature is presented in section 2.3. Additionally, we supplemented measurement details for the friction temperature. The statement “It enables real-time detection and data recording of the surface temperature of objects in the observation area (see Figure 4a).” was added. (Page 6-7, Line 199-209)

  1. Results and discussion - in general, this part of the work is relatively well described

How was figure 9f made? no description what data?

Response: Thank you for your comment. The wear area of Figure 9f is measured by the optical microscope. For intuitive comparison of variation characteristics of wear areas, it is extracted from the macroscopic morphology of wear scar on the rope surface. Additionally, the statement “The wear scar contour map is measured and extracted by the optical microscope. It can be used to more clearly compare changes in the wear area of the wire rope surface.” was added in the revised manuscript to explain the acquisition of the picture. (Page 11, Line 411-413)

  1. Conclusions

Autors said: The bending fatigue life of the wire rope decreases approximately linearly, there is not true; in some areas it may be close to linear and in others to logarithmic

Response: Thank you for your comment. The description in the text is indeed not accurate enough. We modified this sentence. Fatigue life decreases with the increase of sliding contact load. It has been changed to “The bending fatigue life of the wire rope decreases with the increasing sliding contact load.”. (Page 19, Line 658)

Reviewer 3 Report

Wire rope is an important component of crane machinery. Friction and wear behavior of the wire rope in service are directly related to the safe and reliable operation of equipment system and the life safety of operators. Authors studied the tribological behavior of the wire rope was investigated using a homemade rope-pulley sliding friction test rig. However, the current form of this study cannot be acceptable. Some aspects as listed below:

1. In Figure 3, the title of (e) and (f) also should be given. F is the contact load?

2. In Figure 4, (b) should be improved. What is the maximum temperature? (d) more details of the wear parameters should be given.

3. More details about the surface wear mechanism should be given? Besides, the structural parameters of wire rope also should be given.

Author Response

Thank you for your comments. We have studied the comments carefully and have made revisions and modifications which we hope meet with approval. Additionally, the revised portions are marked up using “Track Changes” in revised manuscript. The main corrections in the paper and the responds to the comments are as follows:

  1. In Figure 3, the title of (e) and (f) also should be given. F is the contact load?

Response: Thank you for your advice. the title of (e) and (f) have been given. The statement “(e) Tensioning and fixing of wire rope, (f) bending contact state of wire rope.” was added in Figure 3. (Page 6, Line 190-191) Additionally, F in the Figure is the contact load. We have explained in the revised manuscript. (Page 5, Line 172)

  1. In Figure 4, (b) should be improved. What is the maximum temperature? (d) more details of the wear parameters should be given.

Response: Thank you for your advice. We have modified Figure 4. The maximum temperature refers to the maximum temperature of the sliding contact area of the wire rope. We have changed it to “point of highest temperature”. In Figure 4b, we marked the measuring position of the wear width. (Page 7, Line 221)

  1. More details about the surface wear mechanism should be given? Besides, the structural parameters of wire rope also should be given.

Response: Thank you for your advices. We supplemented the analysis of the wear mechanisms of the wire rope. The detailed descriptions “Because the wire rope is a spiral structure, the sliding contact surface is composed of discontinuous steel wires and rope strands. This causes the friction pair contact sur-face is very uneven. Thus, the sliding resistance is larger and the surface scratches and ploughing effect is more obvious.”, “Therefore, with the increases of contact load, the furrows and spalling pits on the wear surface of the wire rope are significantly reduced. The abrasive wear is reduced during this process. Additionally, As the surface becomes smoother, the wear rate slows down. At the same time, the adhesion is enhanced and the size of the wear de-bris becomes smaller. This also causes the abrasive wear characteristics to be weaken.” and “Furthermore, the morphology characteristics of abrasive wear on wire rope surface are more obvious under smaller contact load.” were added in the revised manuscript. (Page 15, Line 535-539, Line 551-556, Line 557-559)

The structure parameters of the wire rope were added in Table 1. (Page 4, Line 154)

Reviewer 4 Report

Dear, Editor

Tthe paper is so well organized. The experimental work and the results and discussion have high quality or presentation. Moreover, t the friction results and the worn surface morphology also have high quality. It is excellent work for an important application in many places like factories, ports, and other applications. So I accept the paper to be published in the lubricants in its current form

Author Response

Thank you for your comments and support for our paper.

Round 2

Reviewer 1 Report

Thanks to the authors of the article for incorporating my suggested edits. You have now moved the article to a higher level.

I still have the following necessary suggestions for the presented article:

1. Metallographic analysis of steel rope must be included in the article. (determine the exact microstructural composition of the steel).

2. In the text you write that the steel rope is galvanized, it would be appropriate to point out the thickness of this layer when making a metallographic sample.

3. The possibility of supplementing the literature references with, for example, these two articles, which are published by the MDPI publishing house and deal with dry sliding friction depending on different contact tribological pairs (materials), (steel-steel, steel-ceramic ), you can also incorporate these articles into the opening part of your article and point out the change in wear and COF depending on the tribological (friction) pairs.

Krbata, M.; Eckert, M.; Majerik, J.; Barenyi, I. Wear Behavior of High Strength Tool Steel 90MnCrV8 in Contact with Si3N4. Metals 2020, 10, 756. https://doi.org/10.3390/met10060756

Krbata, M.; Eckert, M.; Bartosova, L.; Barenyi, I.; Majerik, J.; Mikuš, P.; Rendkova, P. Dry Sliding Friction of Tool Steels and Their Comparison of Wear in Contact with ZrO2 and X46Cr13. Materials 2020, 13, 2359. https://doi.org/10.3390/ma13102359

4. Please mark all the changes made in yellow, Thank you. After completing these comments, the article can be recommended for publication.

Author Response

  1. Metallographic analysis of steel rope must be included in the article. (determine the exact microstructural composition of the steel).

Response: Thank you for your advice. We have carefully studied the relevant characteristics of the wire rope and communicated carefully with the wire rope manufacturer. We added detailed information about the material properties of the wire rope in the revised manuscript. We emphasize the wire rope production process, the substrate material and density. (Page 4, Line 146, 149-151) Additionally, the surface morphology of steel wires is added in the figure 2f. (Page 4, Line154 and 156) Then, we are very sorry. Because we do not yet have the conditions for metallographic analysis of steel wires, we do not add the relevant content. However, the research carried out in this paper belongs to the category of macro tribology, and more consideration is given to the influence of friction and wear on its fatigue life degradation. Therefore, we believe that the lack of this part of the content will not affect the results and conclusions of this study. Hope you can forgive, thank you.

  1. In the text you write that the steel rope is galvanized, it would be appropriate to point out the thickness of this layer when making a metallographic sample.

Response: Thank you for your advice. We have added the relevant parameters on the thickness of galvanized layer. The statement “Additionally, the thickness of galvanized layer of the steel wire is approximately 7 μm to 20 μm.” was added in the revised manuscript. (Page 4, Line 148-149)

  1. The possibility of supplementing the literature references with, for example, these two articles, which are published by the MDPI publishing house and deal with dry sliding friction depending on different contact tribological pairs (materials), (steel-steel, steel-ceramic), you can also incorporate these articles into the opening part of your article and point out the change in wear and COF depending on the tribological (friction) pairs.

Krbata, M.; Eckert, M.; Majerik, J.; Barenyi, I. Wear Behavior of High Strength Tool Steel 90MnCrV8 in Contact with Si3N4. Metals 2020, 10, 756. https://doi.org/10.3390/met10060756

Krbata, M.; Eckert, M.; Bartosova, L.; Barenyi, I.; Majerik, J.; Mikuš, P.; Rendkova, P. Dry Sliding Friction of Tool Steels and Their Comparison of Wear in Contact with ZrO2 and X46Cr13. Materials 2020, 13, 2359. https://doi.org/10.3390/ma13102359

Response: Thank you for your advice. We have cited these two references to emphasize that different friction pairs have a great influence on the COF and wear characteristics in the test results. In the revised manuscript. (Page 3, Line 121-123)

  1. Please mark all the changes made in yellow, Thank you. After completing these comments, the article can be recommended for publication.

Response: Thank you again for your comments. All modifications in the revised manuscript are marked yellow.

Round 3

Reviewer 1 Report

Thanks to the authors for the answers.

The article in this form is suitable for publication.